# Oral Healthcare Practices and Awareness among the Parents of Autism Spectrum Disorder Children: A Multi-Center Study

**DOI:** 10.3390/children10060978

**Published:** 2023-05-31

**Authors:** Abdullah Saad Alqahtani, Khalid Gufran, Abdulaziz Alsakr, Banna Alnufaiy, Abdulhamid Al Ghwainem, Yasser Mohammed Bin Khames, Rakan Abdullah Althani, Sultan Marshad Almuthaybiri

**Affiliations:** 1Department of Preventive Dental Sciences, College of Dentistry, Prince Sattam Bin Abdulaziz University, Alkharj 11942, Saudi Arabia; ab.alkahtani@psau.edu.sa (A.S.A.); a.alsakr@psau.edu.sa (A.A.); b.alnoufaiy@psau.edu.sa (B.A.); a.alghwainem@psau.edu.sa (A.A.G.); 2College of Dentistry, Prince Sattam Bin Abdulaziz University, Alkharj 11942, Saudi Arabia; 437050302@std.psau.edu.sa (Y.M.B.K.); 437050733@std.psau.edu.sa (R.A.A.); 437050845@std.psau.edu.sa (S.M.A.)

**Keywords:** ASD, autism, parents of an autistic child

## Abstract

This study aimed to evaluate the knowledge and attitudes of the parents of autistic children toward oral health and the practice of oral hygiene habits. A questionnaire was constructed, validated, and distributed to the parents of autistic children at different autistic centers. The questionnaire was divided into three parts: demographic information on the parents of autistic children, the knowledge and attitudes of parents toward oral health, and the practice of oral hygiene habits in the current population. A total of 206 responses were collected. Irrespective of the parents’ educational and employment statuses, most second children were diagnosed with autism, and most of the parents have four or more children. In terms of knowledge and attitudes toward oral health, a total of 90.30% of the patients were aware of the oral healthcare of their child, and 55.80% of parents take them to the dentist for checkups. Moreover, the response to the practice of oral hygiene habits from the parents was positive, and most of the parents assist in the oral hygiene practices of their autistic children. This study showed that most of the parents appeared to have satisfactory knowledge about oral health practices for autistic children. However, additional studies should also be conducted.

## 1. Introduction

Autism or autism spectrum disorder (ASD) is categorized as a neurodevelopmental disorder that has increased in recent years. In the United States, 1 in 40 children is found with this phenotype [1]. Children or adults with ASD are usually characterized by monotonous and stereotyped behavior with less communication and social interaction [2]. Moreover, children with ASD struggle with facial expressions, which are usually different from their peers [3,4,5]. Some children who are diagnosed with ASD can also exhibit a general vocal ability and a normal intelligence quotient (IQ) and oftentimes attend normal school [6]. Even though ASD is a kind of disease with no cure, an early diagnosis of ASD can help develop communication skills [7].

Not all autistic children are characterized by similar behavior. While some children express abnormal linguistic development and impairment of vision and hearing, others develop epilepsy or mental retardation. All these symptoms hinder the oral health of autistic children [8]. Children with ASD also develop harmful oral habits such as tongue-thrusting, lip-biting, and bruxism, to name but a few [9]. A previous study stated that patients with neurodevelopmental diseases tend to show increased gingival diseases and caries [10]. The majority of children with ASD exhibit poor oral and periodontal health, as it requires proper flossing and toothbrushing [10,11,12]. Moreover, because of the sweet tooth of autistic children, an increased incidence of dental caries has been observed among autistic children. Moreover, some medicines for autistic children, such as anticonvulsants and psychostimulants, might cause generalized hyperplastic gingiva [13].

Different psychological well-being levels, along with high levels of stress, have been reported for the parents of autistic children [14,15,16,17,18]. Therefore, scientific guidance, information, and special attention have been provided to the parents of autistic children in many developed countries around the world [19,20]. In order to improve the quality of life for the family members of autistic children, collaboration with professionals who work with patients with ASD is necessary [20]. However, in developing countries, the general knowledge about ASD is not at a satisfactory level compared with developed countries. Hence, autistic children oftentimes do not attain essential dental and medical services [21]. Adequate information about oral care and oral hygiene should be provided to the parents of children with ASD, as poor dental hygiene could disturb speech and mastication, resulting in pain and decreased quality of life and self-confidence [22,23,24]. Since parents are mostly responsible for the overall health issues of autistic children, parents should play an important role in active preemptive dental health. Therefore, parents should be completely educated about the knowledge, attitude, and practices of oral hygiene for autistic children. There have been many studies conducted on the parents of autistic children [20,23,24,25,26,27]; nevertheless, there are insufficient studies on parental knowledge of oral healthcare for autistic children. Therefore, this study aimed to evaluate the knowledge and attitudes of parents of autistic children toward oral health and the practice of oral hygiene habits in the population of the Riyadh region in Saudi Arabia.

## 2. Materials and Methods

The current descriptive cross-sectional study was conducted at the College of Dentistry, Prince Sattam Bin Abdulaziz University. The Standing Committee of Bioethics Research (SCBR) of Prince Sattam Bin Abdulaziz University approved this study protocol (SCBR-016-2023). Moreover, the study was conducted according to the guidelines of the Declaration of Helsinki.

A questionnaire was prepared to distribute to the parents of autistic children. The questionnaire was divided into three parts. The first part contained the demographic information of the parents, with five questions. Part two was confined to nine questions about the knowledge and attitudes of the parents toward oral health. Finally, part three consisted of six questions about the practice of oral hygiene habits. The content of the questionnaire was validated by six professionals from Prince Sattam Bin Abdulaziz University with content and face validity. The content validity ratio was 0.950.

After attaining official approval for the study, authorities at Prince Sattam Bin Abdulaziz University contacted different autistic centers in the Riyadh region for this study. The following autistic centers agreed to participate in this study: Azzam Autism Center, Autism Families Association Al-Kharj, Prince Sultan Center for Support Services for Special Education in Al Rawabi, Autism Families Association Aldwadmi, Prince Sultan Center for Support Services for Special Education in Al Rawabi, Autism Families Association Riyadh, Autism Center of Excellence, and other hospitals in Riyadh city. All the centers provide optimal information about the general and dental health of the children to their parents or guardians once they visited the centers. A validated questionnaire was distributed to the parents of the autistic children in each institute and asked them to fill out the questionnaire and return it the same day. One representative doctor at each institute was responsible for distributing the questionnaire among the parents. If there was any confusion related to any questions, parents could ask the representative doctor for clarification. Each institute sent the questionnaire to Prince Sattam Bin Abdulaziz University. The total time for collecting all the data was three weeks.

The inclusion criteria for participating in this study included parents of autistic children between 3 to 12 years old who were willing to participate in the study. Parents of autistic children who were more than 12 years old and not willing to participate in the study were excluded from the study. Informed consent was obtained from all the parents who agreed to participate in the study. The study was explained to all the parents and they were ensured confidentiality.

### Statistical Analysis

Statistical analyses were performed using the statistical package for Social Science (SPSS) version 27 (IBM, Armonk, NY, USA). Descriptive statistics were used for the frequency distribution of all the responses.

## 3. Results

A total of 206 responses were collected from the parents of autistic children. As this was a questionnaire-based study where parents of autistic children were the main participants, the ages and genders of these autistic children are not reflected in the current outcome. The questionnaire was divided into three parts: demographic information on the parents of autistic children, the knowledge and attitudes of the parents toward oral health, and the practice of oral hygiene habits in the current population. The frequency distribution for the demographic information of the parents showed that 62.60% of the total questionnaire was responded to by mothers. The majority of the parents (40.30%) had a Bachelor’s degree as their level of highest education, and very few parents (2.90%) were illiterate. Irrespective of the education level, a total of 54.40% of parents were working as employees, and 20.40% were housewives. Most parents reported having a second child (31.60%), among other children, who was diagnosed with autism, and the majority of the parents (45.60%) had four or more children. The frequency distribution of the demographic information of the parents is presented in Table 1 and Figure 1.

The frequency distribution for the knowledge and attitudes of the parents of autistic children toward oral health showed that a total of 90.30% and 83% of parents agreed that oral health affects the overall health of the child, and the maintenance of the deciduous teeth is important, respectively. Most of the parents (38.30%) thought children should be taken to the dentist every six months; however, only 12.60% of parents thought children should be taken to the dentist every three months. If the children complain about pain, the majority of the parents (46.10%) take the children to general dentists. Moreover, parents periodically take their children to a general dentist (55.80%) or a general physician (77.20%) for regular checkups. The frequency distribution of the knowledge and attitudes of parents of autistic children toward oral health is shown in Table 2 and Figure 2.

The frequency distribution for the practice of oral hygiene habits showed that toothbrushing with fluoride toothpaste is used as the main tool for maintaining oral health for most of the children (46.60%). Only 10.70% of the children use toothbrushes alone. On a positive note, a total of 50.50% of children brush their teeth twice daily. However, very few patients (18%) use mouthwash in their daily oral hygiene practice. The majority of the autistic children in the study use a regular toothbrush (61.70%) under the supervision of parents (87.90%). Only 12.10% of the parents do not assist their children in maintaining oral hygiene habits. The frequency distribution of the practice of oral hygiene habits is presented in Table 3 and Figure 3.

## 4. Discussion

The current study aimed to evaluate the knowledge and attitudes of the parents of autistic children toward oral health and the practice of oral hygiene habits in the population of the Riyadh region in Saudi Arabia. The number of autistic patients has elevated since the 1980s, probably because of the age of diagnosis, the availability of services, public awareness, and referral patterns [28]. Autism has become a primary concern of public health in numerous countries, and it is characterized by complex behavior with a static brain disorder [29]. Stipulating proper oral care for these types of patients is challenging for parents and requires patience and adequate knowledge. In addition, autistic children are completely dependent on their parents; therefore, parents play an important role in the oral care of autistic children. The current study is a questionnaire-based study, and the questionnaire was constructed and validated for the study. The questionnaire had three sections, including demographic information, knowledge and attitudes, and the practice of oral hygiene habits. All the sections consisted of five to eight questions. The questions were selected and adapted from previous questionnaire studies [21,30].

Learning the demographic information, such as educational levels and socioeconomic status, for the parents of autistic children is important, as it strongly contributes to the overall health not only of autistic children but also of normal children. The mindset of the parents toward a child with ASD will be different if these parents do not have adequate education about the disease. Moreover, people with low socioeconomic statuses oftentimes consider a child with ASD to be a burden and confer less attention to the needs of this child. The current study evaluated the demographic information of the parents of autistic children. It showed that most of the parents achieved a Bachelor’s degree as their level of educational qualification, and the majority of them worked as employees. A previous study assessed the demographic information of the parents of autistic students, and it showed that educational qualifications were similar to the present study [21]. However, employment status was unlike the current study. The majority of parents in the aforementioned study were either businessmen or housewives. Employment status and educational levels are important for taking care of a child with ASD. It is obvious that the general mindset is different from person to person; educated parents can better comprehend autism disease and the importance of taking care of children with ASD. For autistic children who have a linguistic barrier and non-reciprocal behavior, their behavior toward their parents is considered ‘cold’. This causes stress to parents; therefore, parents oftentimes avoid regularly taking care of the autistic child until a serious medical issue occurs [31]. In this case, parents who are educated enough can understand the significance of this disease and the importance of taking care of autistic children. Moreover, employed parents have more financial freedom to provide optimal medical and dental care to autistic children.

On the other hand, second children were most frequently diagnosed as autistic, and the majority of the parents had more than four children in the current population. However, Hajiahmadi et al. [21] showed that the majority of parents have only one child who is diagnosed with ASD. This might be due to the pattern of family planning, which is distinct in different cultures and different countries [32]. However, the number of children is also an imperative issue, as parental attention is divided between children when there are many compared with parents who have only one child. In this study, only 8.70% of parents had one child diagnosed with ASD, while 60% of parents had one autistic child in the studies by Hajiahmadi et al. [21].

The current study showed that 83% of parents believe that deciduous teeth need to be maintained, and 90.30% of parents believe that carious teeth need to be filled. This study included participants who were parents of 3-to-12-year-old children. Therefore, knowledge about maintaining deciduous and mixed dentition is important. Most of the parents showed a positive attitude toward the preservation of deciduous teeth. However, previous studies exhibited that even though parents agreed with the importance of preserving deciduous teeth by filling, lower percentages of parents sought to fill carious teeth [30,33]. There have been many studies conducted on autistic children regarding their need for dental treatment and oral health. Though many studies have stated that the incidence of dental problems significantly increases among autistic children [10,11,12], few studies have shown that there are no significant differences in the incidence of gingivitis, caries, and oral hygiene status compared with healthy children [34]. In addition, some studies have also stated that the prevalence of dental caries is smaller among autistic children compared with their healthy peers [35,36]. However, there are some controversies regarding this statement since autistic children are uncooperative toward dental treatments [13,35]. Therefore, it might be possible that some carious surfaces could go unnoticed during investigations because of this cooperation level [37]. Nevertheless, the current study did not investigate dental caries among autistic children whose parents participated; the majority of the parents are conscious about filling carious teeth in the current population.

In general, it is advisable to visit a dentist every six months for a regular checkup even though this advice is not followed by most people [38]. The current study assessed the frequency of visiting the dentist, and it revealed that only 38.30% of parents take their children to the dentist every six months. Moreover, 32% of parents visited the dentist when their children were in pain. Even though the percentage visiting the dentist every six months is not remarkably higher, the majority of parents at least consider visiting the dentist. The opposite scenario was observed in a previous study, which showed that 61.5% of parents take their children to the dentist when they are in pain, and only 17.3% of parents visit a dentist every six months [30].

Toothbrushing is one of the key components of maintaining good oral health. The current study showed that 50.50% of children brush their teeth twice a day. However, only 17.3% [30] and 14% [21] reported brushing their teeth twice daily in previous studies. The majority of children brush their teeth once a day in these aforementioned studies. However, the current study also showed that a total of 11.70% of children only brush their teeth once a week. In contrast, the majority of parents who participated in the study by Hajiahmadi et al. [21] mentioned that their children brush their teeth occasionally but not every day. Moreover, the frequency of using fluoridated toothpaste has increased in number compared with non-fluoridated toothpaste, which is in line with a previous study [21]. The current study also shows that regular toothbrushing is used by the majority of autistic children, which also supports the frequency outcomes of previous studies [21,30].

ASD is a complicated syndrome where children have different medical issues; therefore, dental health is the least important part for many parents in taking care of them. However, dental care is also important for patients with ASD; therefore, assessing the knowledge and attitudes of the parents of autistic children toward oral care needs to be emphasized. Arranging workshops in order to provide precise information and educate parents about dental care for their autistic children should be directed toward better oral care of affected children.

### Limitations and Strengths of the Study

Even though the frequency of responses received from the parents in the current study was more progressive, the current study only focused on responses from the parents. Conversely, some information about the children, such as age, gender, and educational status, was not recorded. Detailed information about the children could have strengthened the statistical analyses of this study. Moreover, this study was conducted only in the Riyadh region, which is the capital and one of the largest cities in Saudi Arabia; therefore, it is assumed that parents in this region would have the best healthcare support and education for their autistic children. However, it was unclear if the data reflected a representative population or not. A study on other regions could have revealed the knowledge and attitudes of parents regarding dental care; moreover, a comparison could also have been conducted. Moreover, the current study collected all the data based on a convenient sampling method; no sample size or power were calculated. Hence, future studies should be conducted to attain the overall status of the knowledge and attitudes of parents of autistic children toward dental care in this population. However, despite these limitations, the outcome of the current study has some important insights for the social and medical fields. The current study showed that the knowledge and attitudes of the parents of autistic children in the current population are considerably positive. Therefore, optimal social support for the parents of autistic children could help them reach the highest level of knowledge and attitudes in this field. Moreover, medical and dental healthcare providers could endorse a variety of statements regarding the course, treatment, and prognosis of dental issues for parents of autistic children in the current population based on the outcome of this study.

In addition, questionnaires about common dental diseases, such as dental caries, periodontitis, and endodontic and orthodontic treatments for autistic children, as well as the knowledge and attitudes of parents toward advanced dental treatments, could be assessed in the future, which would add meaningful insight to this study.

## 5. Conclusions

This descriptive study showed that most of the parents in the Riyadh region appear to have satisfactory knowledge about oral health practices for autistic children regardless of socio-demographic status. Moreover, most of the parents were aware of how to maintain oral hygiene habits for their autistic children. However, additional studies in other regions of Saudi Arabia should also be conducted.

## Figures and Tables

**Figure 1 children-10-00978-f001:**
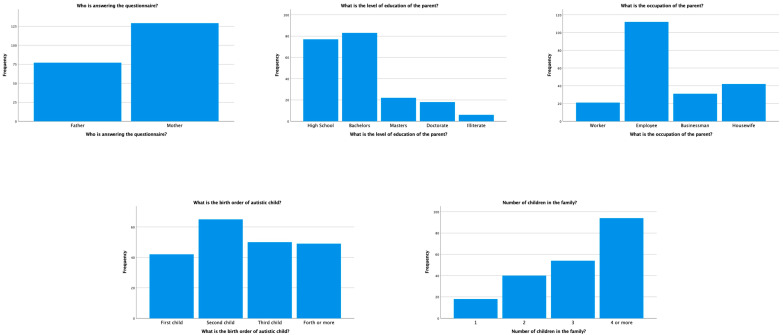
Distribution of the demographic information of parents of autistic children.

**Figure 2 children-10-00978-f002:**
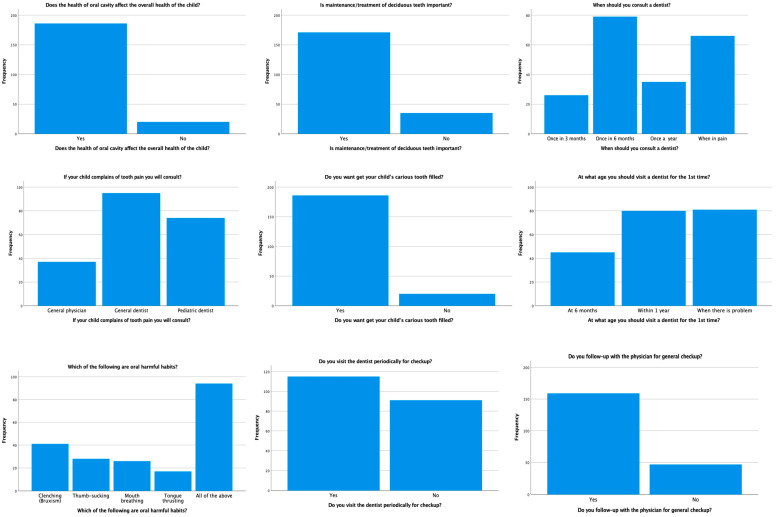
Distribution of knowledge and attitudes of parents toward oral health.

**Figure 3 children-10-00978-f003:**
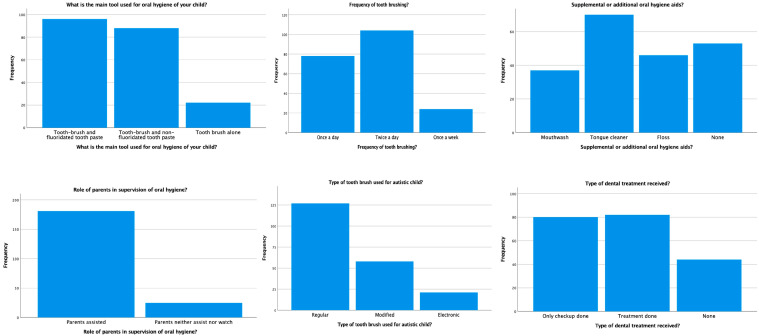
Distribution of the practice of oral hygiene habits.

**Table 1 children-10-00978-t001:** Demographic information of parents of autistic children in the study population.

Questionnaire	Frequency	Percentage (%)
Who is answering the questionnaire?		
Father	77	37.40
Mother	129	62.60
What is the level of education of the parent?		
High school or less	77	37.40
Bachelors	83	40.30
Masters	22	10.70
Doctorate	18	8.70
Illiterate	6	2.90
What is the occupation of the parent?		
Worker	21	10.20
Employee	112	54.40
Businessman	31	15.00
Housewife	42	20.40
What is the birth order of the autistic child?		
First	42	20.40
Second	65	31.60
Third	50	24.30
Fourth or more	49	23.80
Number of children in the family?		
1	18	8.70
2	40	19.40
3	54	26.20
4 or more	94	45.60

**Table 2 children-10-00978-t002:** Knowledge and attitudes of parents toward oral health in the current population.

Questionnaire	Frequency	Percentage (%)
Does the health of the oral cavity affect the overall health of the child?		
Yes	186	90.30
No	20	9.70
Is the maintenance/treatment of deciduous teeth important?		
Yes	171	83.00
No	35	17.00
When should you consult a dentist?		
Once in 3 months	26	12.60
Once in 6 months	79	38.3
Once a year	35	17.00
When in pain	66	32.00
If your child complains of tooth pain, whom do you consult?		
General Physician	37	18.00
General Dentist	95	46.10
Pediatric Dentist	74	35.90
Do you want to have your child’s carious tooth filled?		
Yes	186	90.30
No	20	9.70
At what age should you visit a dentist for the 1st time?		
At 6 months	45	21.80
Within 1 year	80	38.80
When there is a problem	81	39.30
Which of the following are harmful oral habits?		
Clenching (bruxism)	41	19.90
Thumb-sucking	28	13.60
Mouth breathing	26	12.60
Tongue-thrusting	17	8.30
All of the above	94	45.60
Do you visit the dentist periodically for a checkup?		
Yes	115	55.80
No	91	44.20
Do you follow up with the physician for a general checkup?		
Yes	159	77.20
No	47	22.80

**Table 3 children-10-00978-t003:** The practice of oral hygiene habits in the current population.

Questionnaire	Frequency	Percentage (%)
What is the main tool used for the oral hygiene of your child?		
Toothbrush and fluoridated toothpaste	96	46.60
Toothbrush and non-fluoridated toothpaste	88	42.70
Toothbrush alone	22	10.70
What is the frequency of tooth brushing?		
Once a day	78	37.90
Twice a day	104	50.50
Once a week	24	11.70
Supplemental or additional oral hygiene aids?		
Mouthwash	37	18.00
Tongue cleaner	70	34.00
Floss	46	22.30
None	53	25.70
Role of parents in the supervision of oral hygiene?		
Parents assist	181	87.90
Parents neither assist nor watch	25	12.10
What type of toothbrush is used for autistic children?		
Regular	127	61.70
Modified	58	28.20
Electronic	21	10.20
Type of dental treatment received?		
Only checkup done	80	38.80
Treatment done	82	39.80
None	44	21.40

## Data Availability

Not applicable.

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
