# Peer review of "Oral Healthcare Practices and Awareness among the Parents of Autism Spectrum Disorder Children: A Multi-Center Study"

_children, 2023, doi:10.3390/children10060978_

Round 1
Reviewer 1 Report
1. What is the future continuation of this study?
2. Data can be presented in graphical form for better understanding.
3. Addition of common dental diseases questionnaire about Dental caries, and periodontitis in this survey could be more attractive.
4. A graphical abstract could be more eye-catching for the reviewer.
1. What is the future continuation of this study?
2. Data can be presented in graphical form for better understanding.
3. Addition of common dental diseases questionnaire about Dental caries, and periodontitis in this survey could be more attractive.
4. A graphical abstract could be more eye-catching for the reviewer.
Author Response
"Please see the attachment."
Comments and Suggestions for Authors,
1. What is the future continuation of this study?
Reply to reviewer: Thank you so much for your comment. Future continuation of this study has been added.
2. Data can be presented in graphical form for better understanding.
Reply to reviewer: Thank you so much for your comment. Figures have been added as per comment.
3. Addition of common dental diseases questionnaire about Dental caries, and periodontitis in this survey could be more attractive.
Reply to reviewer: Thank you so much for your comments and suggestions. All authors agreed that addition of common dental diseases questionnaire about Dental caries, and periodontitis in this survey could be more attractive. However, timeline for this research has been accomplished; therefore, addition of a new data would be difficult at this point.
4. A graphical abstract could be more eye-catching for the reviewer.
Reply to reviewer: Thank you so much for your comments and suggestions. We agree that a graphical abstract could be more eye catching; however, the authors decided to keep the abstract in traditional format.

Reviewer 2 Report
Dear authors, it has been a pleasure for me to have reviewed this article, not only because of my clinical, teaching and research experience with patients with special needs, but also because of the respect and affection I have for you.
I am very pleased that research is being done on this topic and I congratulate you for the research work done. However, I believe that the following suggestions would help increase the quality of your article and improve its comprehension by the readers.
ABSTRACT
- Lines 19, 20 and 21: In terms of knowledge and attitude towards oral health, most of the patients were aware of the oral health care of their child and regularly take them to the dentist for checkups. Moreover, the response to the practice of oral hygiene habits from the parents is in- conceivable and most of the parents assisted in the oral hygiene practice of their autistic child.
As this is a brief summary of the results obtained, please replace the terms "most the patients" and “regularly” with numerical data.
INTRODUCTION
-Bartolomé-Villar, B.; Mourelle-Martínez, M.R.; Diéguez-Pérez, M.; de Nova-García, M.J. Incidence of oral health in paediatric patients with disabilities: Sensory disorders and autism spectrum disorder. Systematic review II. J Clin Exp Dent 2016, 8, e344-51.
Please add information from this article, I think it can enrich the introduction.
MATERIALS AND METHODS
- Lines 73 and 74: This study aimed to evaluate the knowledge, attitude, and oral health practice of the parents of autistic children and their demographical information in the Riyadh region.
It is advisable not to repeat the objective of the study in this section, as it has already been reflected in the introduction.
- Please answer the following questions in this section:
Time taken from dissemination of the questionnaire to obtaining all responses?
Regarding the systematics of the interview with the parents: How was the questionnaire carried out (online, face-to-face...), by whom? How much time did each parent have to answer?
¿The sample studied is representative of the population?
Have you found the sample power? If so, can you reflect it?
- Line 96: The subtitle does not follow the regulations.
Change Statiscal analysis to 2.1. Statiscal analysis
RESULTS
Have the results taken into account the age and sex of the children?
DISCUSSION
-Lines 151-153: Demographic information for the parents of autistic children is important as it con-150 tributes major roles in the overall health not only of autistic children but also of normal 151 children.
Please explain the basis for this statement.
Explain the basis for the selection of the questions asked.
Based on the results obtained, indicate what application your study has in the social and medical fields.
Add the subtitle “limitations and strengths of the study” and develop its content.
COCLUSIONS
-In this section they only talk about knowledge without mentioning attitudes.
Please add additional information.
REFERENCES
For the benefit of the reader, please add in this section a link to [PubMed] and/or [CrossRef] for faster location of articles in the scientific literature.
Author Response
"Please see the attachment."

Round 2
Reviewer 2 Report
Dear authors, thank you for the changes made to the article.
However, no information has been reflected on some of the suggestions regarding the fact that the sample is representative or not of the general population and on the sample power. Please refer to these two aspects, either in the methodology section or in the limitations section if you consider this to be a limitation.
Author Response
Thank you so much for your comments. I appreciate your feedback.
As per your suggestions, the missing information has been added to the limitations of the study.